Review Article

# Revisiting giant virus-host dynamics in brown algae: old stories and new perspectives

Carole Duchêne[1,2], Liping Wang[1,2] & Susana M Coelho [1✉]

## Abstract

The recent discovery of widespread giant virus sequences integrated into the genomes of diverse eukaryotes, and in particular marine lineages, has reignited interest in the molecular mechanisms underlying giant virus–host interactions. The brown alga *Ectocarpus* represents a compelling and historically rich model for such studies. As early as the 1970s, it was used to investigate latent infections by giant double-stranded DNA viruses, with elegant classical genetics and electron microscopy approaches revealing key aspects of virus–host life cycle coordination. However, progress was limited by the lack of molecular and genomic tools. In this review, we revisit these foundational studies through the lens of recent technological advances, including the development of genetic and genomic resources for brown algae. These tools now enable mechanistic insights into giant viral integration, latency, activation and host response. We highlight how *Ectocarpus* and related systems can illuminate both the evolutionary and ecological dimensions of virus–host dynamics, with a particular emphasis on the molecular and genetic mechanisms that mediate these complex interactions.

**Keywords** Brown Algae; Viruses; Nucleocytoviricota
**Subject Categories** Evolution & Ecology; Microbiology, Virology & Host Pathogen Interaction

## Introduction

Viruses are now recognized as the most abundant biological entities in marine habitats, a finding that has reshaped our view of ocean ecosystems and their complexity (Suttle, 2005). Among them, giant viruses of the phylum *Nucleocytoviricota* have emerged as major players in eukaryotic biology, with an expanding diversity that continues to redefine virus–host relationships across the tree of life (Forterre and Gaïa, 2016). Marine eukaryotes serve as important hosts for these viruses, and their infections can strongly influence microbe population dynamics and phenomena such as algal blooms (Vincent and Vardi, 2023). In particular, viral lysis of phytoplankton not only terminates blooms by directly reducing host cell abundance but also releases dissolved organic matter that fuels microbial loop processes and alters nutrient cycling (Hevroni et al, 2023). In some cases, viral activity can control the timing, magnitude, and succession of blooms, shaping community structure at ecosystem scales. Moreover, interactions between viruses, algal hosts, and other microbial partners can have cascading effects on biogeochemical fluxes, including carbon export and oxygen production in the ocean (Mojica and Brussaard, 2025). Collectively, these observations highlight marine viruses as key regulators of ocean ecosystem dynamics and biogeochemical fluxes. Within this broader framework, phaeo-viruses, large double-stranded DNA viruses of the phylum *Nucleocytoviricota*, represent a particularly intriguing case within marine virus–host interactions. These viruses infect multicellular brown algae (Box 1), establishing persistent infections that can be transmitted through the host's reproductive life cycle. Early studies, particularly those conducted in the 1990s, suggested a latent life cycle with integration of viral DNA into the host genome and controlled expression of viral genes (Fig. 1). Classical approaches, combining genetics, cytology, and electron micro-scopy provided key insights into how these viruses maintain long-term infections without immediately lysing their hosts. These foundational studies established a strong conceptual framework, demonstrating that these viruses could serve as models for understanding virus–host coordination in complex multicellular eukaryotes. Despite this early promise, progress in the field stalled over the past decades. The lack of molecular and genomic tools limited mechanistic studies, and consequently, the biology of phaeoviruses remained poorly understood relative to other giant viruses.

This situation is changing rapidly. Recent advances in brown algal genomics and molecular biology provide an opportunity to revisit earlier studies. With modern tools, we can now dissect the molecular mechanisms of phaeovirus integration, latency, activation, and host response, paving the way for a deeper understanding of how these viruses influence host biology and ecosystem dynamics. This review places these pioneering studies in the context of emerging molecular insights, highlighting the contributions of new tools for brown algae research, particularly in the model organism *Ectocarpus*.

[1]Max Planck Institute for Biology Tübingen, 72076 Tübingen, Germany. [2]These authors contributed equally: Carole Duchêne, Liping Wang.
✉E-mail: susana.coelho@tuebingen.mpg.de

    

**Glossary**

| | | | |
|---|---|---|---|
| Chronic infection | persistent infection in which viral particles are continuously produced | Latent infection | rise to the male and female gametes. persistent infections in which complete copies of the viral genome persist in the host for extended periods without the continuous production of infectious virus. |
| EVE (Endogenous Viral Element) | genetic element in a host genome derived from a virus sequence. EVEs can represent full-length viruses or fragments of viral genomes, and can derive from active integration mechanisms or accidental endogenization of a replicating virus. | Lytic infection | active replication of the viral genome, characterized by the release of new progeny virus particles, often upon the lysis of the host cell. |
| Gametophyte | One of the two generations (usually haploid and multicellular) in plants and algae that undergo alternation of generations. Here, gametophytes develop from spores and give | Sporophyte | One of the two generations (usually diploid and multicellular) in the life cycle of plants and algae that undergo alternation of generations. Here, the sporophyte produces haploid spores |

# Giant viruses infecting algae

*Nucleocytoviricota*, formerly referred to as the nucleocytoplasmic large DNA viruses (NCLDV), constitute a distinct viral phylum of double-stranded DNA viruses with exceptionally large genomes, in some cases exceeding 2.5 Mb (Philippe et al, 2013; Fischer, 2016). Members of this group display remarkable host diversity, infecting organisms ranging from mammals to microalgae, and encompassing the vast phylogenetic breadth of protists. Notably, *Acanthamoeba*, the host from which the first giant virus (mimivirus) was isolated, has become emblematic in the study of this lineage (Raoult et al, 2004). In addition to these well-characterized systems, a multitude of yet-unidentified hosts are thought to sustain the extensive diversity of *Nucleocytoviricota* (Aylward et al, 2021; Schulz et al, 2022).

Historically, double-stranded DNA viruses infecting algae were classified within the *Phycodnaviridae* family, a grouping based primarily by pioneering studies of eukaryotic algae from both marine and freshwater environments (Wilson et al, 2009). Representative examples include the green alga *Chlorella*, targeted by chloroviruses such as Paramecium bursaria chlorella virus (PBCV), and brown algae (Phaeophyceae), which host phaeoviruses such as Ectocarpus siliculosus virus-1 (EsV-1). These viruses were initially considered to form a monophyletic order within *Nucleocytoviricota*. However, the identification of novel virus–host associations, coupled with insights from environmental metaomics, has fundamentally challenged this view (Aylward et al, 2021). Indeed, recent phylogenetic analyses, including metagenome-assembled viral genomes, place algae-infecting viruses in different orders of *Nucleocytoviricota*: coccolithoviruses (infecting the haptophyte *Emiliana huxleii*) and phaeoviruses are part of the proposed '*Pandoravirales*' order, prasinoviruses infecting green algae in the *Algavirales* order, while viruses infecting diverse algae such as the stramenopile *Aureococcus*, the green algae *Tetraselmis* and the haptophyte *Prymnesium kappa* are classified in different families of the *Imitervirales* (Aylward et al, 2021; Koonin et al, 2024) (Fig. 2). Beyond their genetic diversity, these viruses exert profound ecological impacts by modulating algal population dynamics, driving bloom termination, and shaping nutrient cycling and carbon export in aquatic ecosystems (Mojica and Brussaard, 2025). It is perhaps unsurprising that the diversity of viruses infecting algae mirrors the evolutionary breadth of their hosts, as algae represent a deeply divergent assemblage of eukaryotes with complex evolutionary histories. Notably, even within the same algal genus, distinct giant viruses from different clades have been shown

to establish infections (e.g., Prymnesium kappa viruses and Haptolina ericina virus infected the same hosts (Johannessen et al, 2015)). A substantial fraction of viral clades, however, remain without identified hosts. Among more than 2500 *Nucleocytoviricota* genomes sequenced to date, only around 200 originate from cultivated isolates with known hosts (Schulz et al, 2022). Emerging methodologies are beginning to address this gap, including approaches based on co-occurrence and interaction networks (e.g., Endo et al, 2020), single-cell genomics (e.g., Needham et al, 2019), and more recently, single-cell transcriptomics (e.g., Fromm et al, 2024) and chromosome conformation capture (Hi-C) techniques (e.g., Bignaud et al, 2025). Together, these approaches hold promise for resolving previously unknown host–virus associations.

The viruses known to infect brown algae (Phaeophyceae) form a monophyletic lineage, the phaeoviruses, within the proposed order '*Pandoravirales*'. Their closest characterized relative is coccolithoviruses, e.g., the Emiliania huxleyi virus (EhuxV), a central model in marine virology (Ku et al, 2020; Kuhlisch et al, 2021; Joffe et al, 2024; Shaler et al, 2025). Despite their ecological and evolutionary significance, the biology of phaeoviruses remains poorly understood, with most available knowledge stemming from the pioneering studies of Dieter Müller and colleagues during the 1990s and early 2000s (Fig. 1A). Their unusual infection strategies and complex interactions with algal hosts (Müller et al, 1998; Schroeder, 2015) suggest that they represent an especially compelling system for future study of host–virus interactions and latent life cycles. Among them, EsV-1 has emerged as the best-studied representative, providing a foundation for our current understanding of phaeovirus biology.

## Historical foundations: the *Ectocarpus*-phaeovirus system

Viral symptoms in *Ectocarpus* were described almost simultaneously with the discovery of the alga itself. In the 1890s, as phycologists documented brown algal diversity across Europe, the French phycologist Camille Sauvageau reported abnormal reproductive structures and proposed that these abnormalities were caused by an unidentified parasite (Sauvageau, 1896) (Fig. 1A). Nearly 80 years later, virus-like particles were first visualized by electron microscopy in the reproductive organs of *Ectocarpus* (Clitheroe and Evans, 1974). Foundational work by Dieter Müller and colleagues in the 1990s subsequently elucidated key aspects of phaeovirus biology. In their initial descriptions, Müller and collaborators demonstrated that viral symptoms could be

## Box 1   What are brown algae?

Brown algae (Phaeophyceae) represent a highly diverse lineage of Stramenopiles that has independently evolved complex multicellularity. Their wide range of morphologies, life cycles, and reproductive strategies means that many species within this group have strong potential to serve as model systems for addressing fundamental biological questions. Established models include the fucoids, which have long been used for cell biology studies (Coelho et al, 2002; Coelho and Cock, 2020), and *Ectocarpus* sp., which is increasingly employed for molecular genetics and developmental research. More recently, kelps such as *Saccharina* and *Macrocystis* have emerged as promising systems for studying tissue differentiation, large-scale thallus morphogenesis, and ecological adaptations. Together, these species provide opportunities to investigate development, reproduction, and evolution across the brown algal lineage, while also offering insights relevant for biotechnology and sustainable resource applications.

Brown algae as comparative models for multicellularity and development. Brown algae (Phaeophyceae) are multicellular Stramenopiles, evolutionarily distinct from land plants, red algae, and opisthokonts. They display a broad range of morphologies, from simple filaments (*Ectocarpus* sp.) to large parenchymatous thalli with multiple cell types (kelps, Laminariales), and exhibit diverse reproductive strategies and sexual systems (Batista et al, 2024). Their independent evolution of complex multicellularity and unique developmental programs makes them valuable comparative models for understanding the origins and mechanisms of multicellularity, tissue differentiation, and life-cycle regulation. Established models include fucoids (*Fucus* sp.) for cell biology, *Ectocarpus* sp. for molecular genetics, and kelps for morphogenesis and ecological studies, providing complementary systems to study fundamental biological processes in a lineage distant from plants and animals.

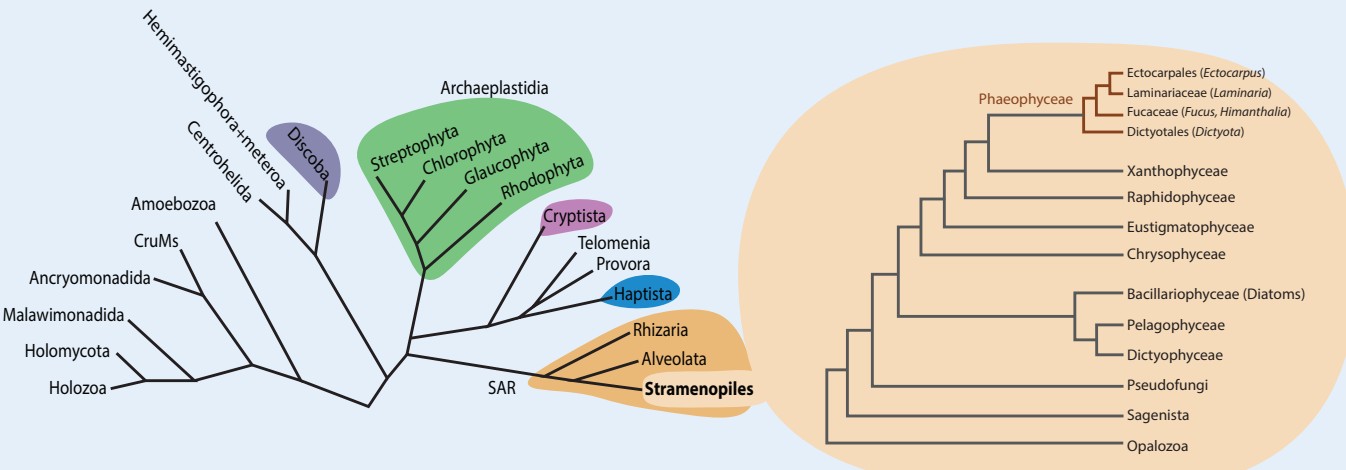

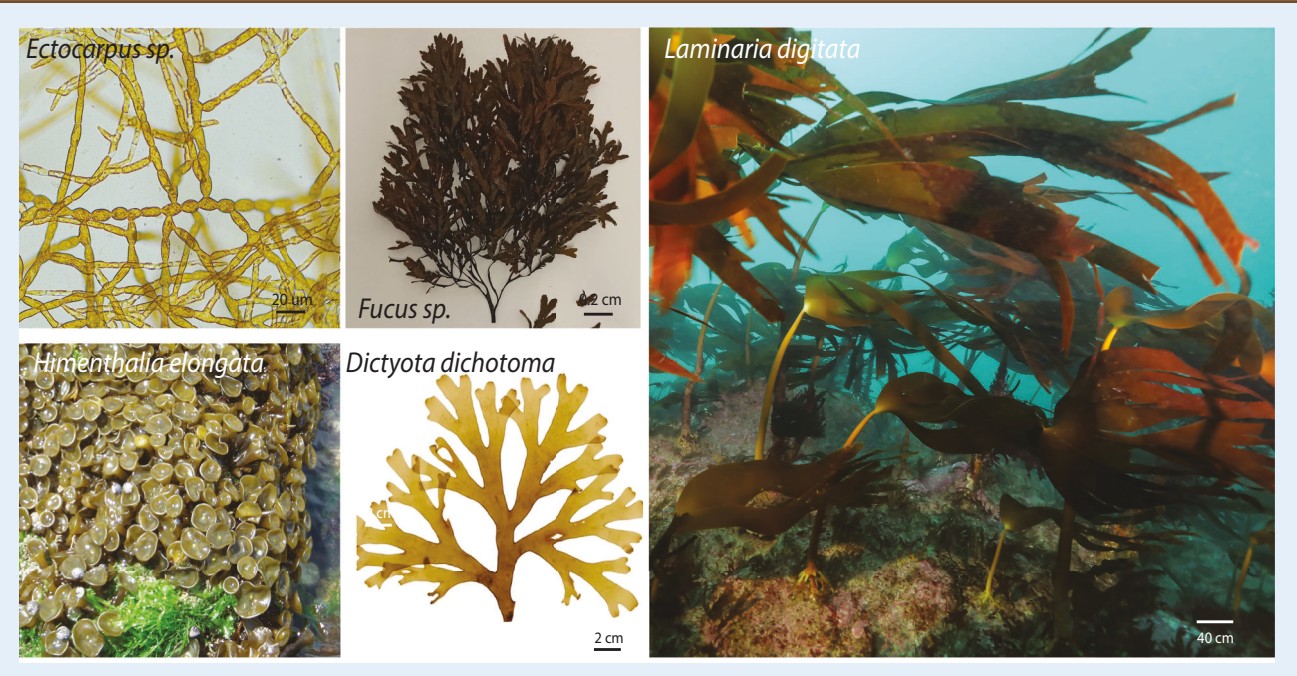

**Figure Box 1**. The diversity of Eukaryotic algae: schematic tree of Eukaryotes, adapted from Williamson et al, 2025, and expanded Stramenopile branch to illustrate the position of Phaeophyceae (adapted from Cho et al, 2024; Denoeud et al, 2024; Terpis et al, 2025). Lineages containing eukaryotic algae are highlighted in color. Photos kindly provided by Kenny Bogaert, Rémy Luthringer and Wilfried Thomas (CNRS).

## A

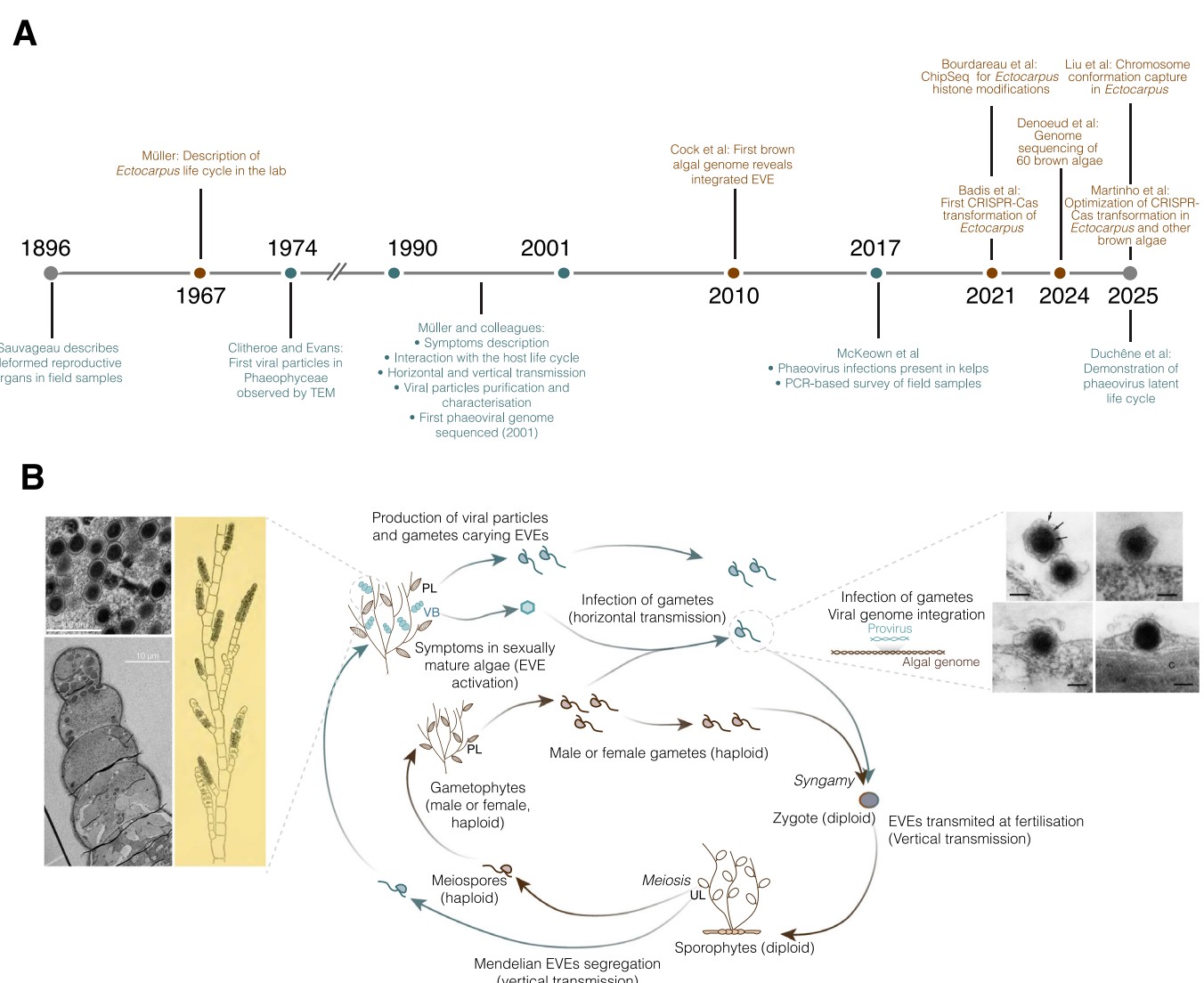

**Figure 1. The *Ectocarpus*–phaeovirus system: from historical observations to modern tools.**

**(A)** Timeline summarizing the history of *Ectocarpus*–phaeovirus research. Top: Important milestones in *Ectocarpus* research (in brown). Bottom: Discovery and foundation work on phaeoviruses (in blue). **(B)** Simplified diagram of *Ectocarpus* life cycle showing the stages where viral activation occurs. The brown inner arrows depict the life cycle for *Ectocarpus* with no virus infection, whereas the blue outer arrows illustrate how the virus manipulates the host life cycle. The algal sporophyte (diploid) produces meiospores via meiosis in the unilocular sporangia (UL). The (haploid) meiospores inherit EVEs present in the sporophyte genome in a Mendelian manner (horizontal transmission of the latent EVE). They develop as male and female gametophytes, which in turn produce gametes by mitosis in reproductive structures called plurilocular sporangia (PL). If the meiospore has inherited an active EVE, it can develop viral symptoms in plurilocular sporangia in the form of viral bladders (VB). Theses viral bladders were first described in 1896 by Sauvageau (see drawing on the left, from Sauvageau (1896)), and consist of swollen cells filled with viral particles (Transmission Electron Microscopy (TEM) images on the left). Male and female gametophytes release male and female gametes, while actively infected individual release viral particles. These particles can infect naïve, cell-wall-free, swimming gametes, by attaching to their cell surface (Right; TEM images from (Maier et al, 2002). The viral genome is then released into the gametes and transferred to the nucleus, where it integrates into the algal genome. Note that gametophytes infected by the virus may also produce healthy gametes, that carry the integrated EVE in their genomes. These gametes then fuse with gametes from the other sex, giving rise to a new diploid zygote, transmitting the latent virus to the next generation.

transmitted horizontally via infection of zoids (Müller et al, 1990). In addition to horizontal transmission, viral symptoms were also observed to persist through mitotic divisions in algal cultures, and to be inherited vertically throughout the algal life cycle, both at fertilization and during meiosis. Strikingly, the symptoms segregated in a Mendelian fashion, leading to the early hypothesis that the viral genome existed in a proviral form, closely associated with, or integrated into one of the host autosomes (Müller, 1991).

Subsequent studies from the same laboratory (e.g., Delaroque et al, 1999; Delaroque and Boland, 2008), together with more recent work (Duchêne et al, 2025, see "The genomic era" section below), confirmed this hypothesis, establishing that the viral genome is maintained as an endogenous viral element (EVE) inserted into the algal genome.

*Ectocarpus* alternates between haploid (gametophyte) and diploid (sporophyte) stages, both of which develop from free-

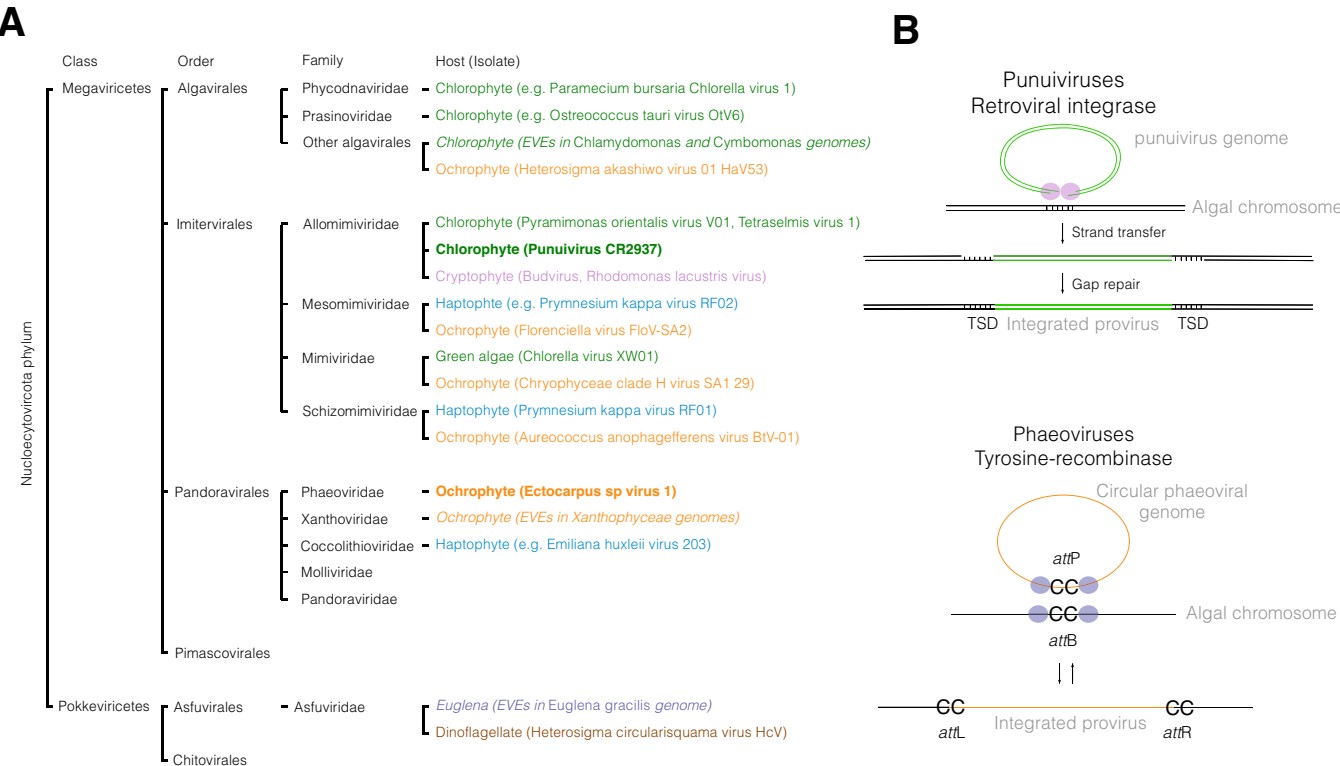

**Figure 2. *Nucleocytoviricota* classification and mechanisms of integration.**

(**A**) Schematic tree of *Nucleocytoviricota* classification, highlighting viruses infecting algae. In italics are viruses known only from EVEs (Endogenous Viral Elements) in their host genome, while the bold highlight the two viral lineages for which a latent life cycle is demonstrated (phaeoviruses) or strongly suspected (punuiviruses). Tree was adapted from (Aylward et al, 2021; Koonin et al, 2024), with additional lineages from (Nagasaki et al, 2003; Ogata et al, 2009; Moniruzzaman et al, 2022; Sheng et al, 2022; Aylward et al, 2023; Thomy et al, 2024; Vieira et al, 2024; Byl et al, 2025; Jivaji et al, 2025). (**B**) Proposed mechanisms for integration of viruses with latent integrated life cycle. Punuiviruses integration takes place via retroviral integrases (see text), leaving Target Site Duplications (TSD), while phaeoviruses integrate via a tyrosine recombinase. For simplicity and to explain the formation of the TSD, only the retroviral mechanisms is represented as double-stranded DNA.

living multicellular stages (Fig. 1B). Infectious particles can be observed in cultures at specific developmental transitions, particularly during gametogenesis and sporogenesis. Environmental factors, such as temperature decreases (from standard growth conditions at 14 °C to 10 °C) increases the severity of viral symptoms, with a higher proportion of reproductive structures producing viral particles (Müller, 1991; Lanka et al, 1993). The released particles are capable of infecting naive *Ectocarpus* wall-less zoids (Müller et al, 1990; Klein et al, 1995, Fig. 1B). Upon infection, the virus attaches to the target cell using a yet unknown mechanism and releases its viral core into the cytosol. The viral core migrate to the nucleus, where the viral DNA integrates into the host genome, completing the infectious cycle (Maier et al, 2002).

The *Ectocarpus*–EsV-1 system exemplifies a lysogenic viral strategy: long-term maintenance as a heritable, largely "silent" EVE integrated into the algal genome, punctuated by environmentally or developmentally triggered reactivation that restores a lytic phase. At the time of its discovery, however, there was no direct proof linking the integrated viral element with either the appearance of symptoms nor the production of viral particles. Conclusive evidence for this connection only came to light more than 30 years later (Duchêne et al, 2025, see "The genomic era" section below).

In addition to documenting symptom transmission, Dieter Müller's laboratory also characterized phaeovirus virions, investigating particle morphology and, notably, beginning to analyze individual viral proteins (e.g., GP1, VHK (Klein et al, 1995; Delaroque et al, 2000a; Delaroque et al, 2000b)) well before the advent of modern proteomics. The group further provided the first descriptions of the viral genome (Lanka et al, 1993; Lee et al, 1995), work that culminated in the full-length sequencing of the EsV-1 genome in 2001 (Delaroque et al, 2001). Importantly, Müller's research was not confined to a single *Ectocarpus* strain in culture but extended across diverse *Ectocarpus* isolates and other brown algal species from different geographical origins, thereby demonstrating the global prevalence of these viruses in Phaeophyceae (Müller and Stache, 1992; Sengco et al, 1996). Comparable symptoms and associated virus-like particles were later reported in other brown algae (Kapp et al, 1997; McKeown et al, 2017), though never with the same level of detail achieved in the *Ectocarpus*–EsV-1 system, which subsequently emerged as the model for phaeovirus biology.

Following this pioneering phase, progress slowed considerably after the early 2000s, largely due to the absence of genetic tools for studying brown algal hosts, which severely constrained mechanistic research on phaeoviruses (see Fig. 1A). Only sporadic studies,

primarily addressing ecological aspects, were published between 2000 and 2020 (e.g., McKeown et al, 2018; see ecology section below). The field was revitalized, however, with the discovery that phaeoviruses are not only globally distributed but also abundant as integrated elements within brown algal genomes (Denoeud et al, 2024), placing them back at the forefront of marine virology.

## The genomic era: hidden giant viruses in brown algal host genomes

Recent advances in sequencing technologies revealed the breadth and the ubiquity of viral elements integrated in eukaryotic genomes. With more and more (good quality) eukaryotic genomes becoming available, we are only starting to realize the diversity of *Nucleocytoviricota* and their hosts and how these interactions shaped the evolution of eukaryotes (Gallot-Lavallée and Blanc, 2017; Moniruzzaman et al, 2022; Zhao et al, 2023; Sarre et al, 2024; Jivaji et al, 2025). However, if the detection of EVEs in a genome can inform on past encounters between a host and its viruses, the physiological and evolutionary meaning of these insertions is not always evident.

The *Ectocarpus* genome, and the Phaeophyceae-phaeovirus system in general, exemplifies the relationship between viral insertion and viral symptoms, as well as the dynamics of these insertions across evolutionary times. The first *Ectocarpus* genome to be sequenced was found to host several virus pieces, including the insertion of a putatively full-length phaeovirus closely related to EsV-1 (Cock et al, 2010). However, this *Ectocarpus* strain (species 7, strain Ec32) is not the same strain as the one used in the historically studies on EsV-1, and never showed symptoms in culture. The *Ectocarpus* strain hosting EsV-1 was sequenced only later, along with other *Ectocarpus* strains showing symptoms, allowing the definitive demonstration that phaeoviral EVEs are reactivating to cause symptoms using classical and reverse genetics (Duchêne et al, 2025). These genomes host one or several active EVEs (i.e., able to cause symptoms), along with seemingly full-length EVEs not associated with symptoms and smaller EVE pieces. The later are likely the result of EVE degradations or host defenses against invading nucleic acids during evolution, and might explain the first reports that *Ectocarpus* genome contains several viral DNA pieces (Delaroque and Boland, 2008).

Recent large-scale comparative genomic studies have revealed that viral integrations are a pervasive feature of brown algal genomes. Large fragments of giant double-stranded DNA viruses related to phaeoviruses have been detected across diverse brown algal lineages (Cock et al, 2010; Denoeud et al, 2024; Mckeown et al, 2025). A recent comprehensive analysis of dozens of newly sequenced genomes demonstrated that viral element insertions are not restricted to isolated taxa but are distributed throughout the brown algal phylogeny (Denoeud et al, 2024). This phylogenetic distribution suggests that brown alga-virus interactions have been recurrent over evolutionary timescales, leaving behind a substantial genomic signature. These genomic findings corroborate previous PCR-based surveys of natural populations that suggested widespread occurrence of viral sequences in algal communities (e.g., McKeown et al, 2018).

The viral sequences detected often comprise large portions of phaeoviral genomes, in some cases approaching near-complete genomes rather than short remnants. This observation aligns with the known biology of phaeoviruses, which integrate into the genomes of their hosts and can establish long-lasting, latent infections (Duchêne et al, 2025). While many of the integrated sequences identified are fragmented and degraded, their extent suggests that past infections involved major viral contributions to host genomes. These viral regions also include fragments with affinities to more distantly related nucleocytoviruses (*Imitervirales*) (Denoeud et al, 2024). These results reinforce the idea that phaeoviruses, and possibly other nucleocytoviruses, have not only shaped brown algal biology through infection cycles but have also contributed to the genomic architecture of their hosts and helped shaping their evolution.

From an evolutionary perspective, the widespread presence of viral genome fragments points to repeated episodes of endogenization during the diversification of brown algae. The diversity of phaeoviral EVEs and their non-congruence with brown algal phylogeny suggest repeated event of infection rather than a single event followed by duplication and diversification, but this remains to be further studied. Such events may have contributed to changes in brown algal genome size and structure, and potentially offered raw genetic material for functional innovation through the domestication of viral genes (Macaisne et al, 2017; Denoeud et al, 2024). In other words, although the activity and biological significance of most integrated elements remain to be determined, their ubiquity suggests that interactions with giant viruses have been a defining feature in the evolutionary history of brown algae (see "Ecological and evolutionary implications" section).

Together, these findings open new avenues for exploring how host–virus relationships extend beyond infection dynamics to leave lasting imprints on genome evolution. A striking example of this phenomenon comes from the filamentous brown alga *Porterinema*, whose genome was recently shown to contain multiple, diverse EVEs (Mckeown et al, 2025). The diversity and abundance of these viral insertions point to a history of repeated, independent integration events rather than a single viral endogenization followed by vertical transmission. In several cases, the integrated elements span extensive genomic regions, retaining gene repertoires typical of giant dsDNA viruses, including those involved in replication and virion structure. The *Porterinema* case study highlights the potential evolutionary and functional implications of viral endogenization. Some of the endogenous viral fragments appear relatively intact and may retain transcriptional activity, suggesting possible roles in host biology or viral reactivation. Others are highly degraded, consistent with ancient and non-functional remnants. Together, these findings demonstrate how repeated interactions with giant viruses have left a complex genomic legacy in *Porterinema*. Beyond documenting viral persistence, they emphasize the potential of endogenized elements to shape genome architecture, act as a source of genetic novelty, and provide a molecular fingerprint of long-term co-evolution between brown algae and their viruses.

## Mechanisms of integration, latency, and activation

Phaeoviruses exhibit an integration strategy that differentiates them from many other nucleocytoviruses. Unlike typical giant viruses such as mimiviruses, which mainly employ lytic replication strategies (Scola et al, 2003; Machado et al, 2022), phaeoviruses

establish long-term latent infections by inserting their DNA into the nuclei of reproductive zoids in brown algae (Duchêne et al, 2025). This integration ensures vertical transmission of the viral genome through the germline, allowing the virus to persist for multiple generations (Müller, 1991; Duchêne et al, 2025). In contrast, other nucleocytoviruses generally leave minimal or no genomic traces in their hosts, as their life cycles rely on repeated horizontal transmission and lytic replication (Abergel and Claverie, 2020).

The only established Nucleocytoviricota model for which a latent life cycle is demonstrated are phaeoviruses (Duchêne et al, 2025), while in punuiviruses (Imitervirales, see Fig. 2A, Erazo-Garcia et al, 2025) a cryptic infection has been described. Some genomic evidence suggests that Xanthoviridae, infecting xanthophyte algae (from the proposed order 'Pandoravirales', Duchêne et al, 2025) and Mycodnaviridae, infecting fungi (Imitervirales, Myers et al, 2025) could also use a lysogenic life cycle.

Phaeoviruses are thought to integrate by site-specific recombination between a site in the algae genome, termed attB following phage nomenclature, and a site in the viral genome (termed attP), giving rise to the inserted virus flanked by attR and attL sites (Duchêne et al, 2025). The attP site is conserved among phaeoviruses, whereas the attB site contains only a short conserved CC motif, which is also present in attP (Fig. 2B). Note that the full diversity of phaeoviruses has not yet been explored. For example, viruses infecting the brown alga Feldmannia possss different attB/attP sequences, and integration occurs at CG sites (Meints et al, 2008). However, despite sequence divergence, all phaeoviruses including the Feldmannia viruses presumably integrate via a conserved mechanism. In both type of attP site, the insertion site (CC or CG) is flanked by inverted repeats that could lead to the formation of a hairpin exposing the CC/CG nucleotides. The integration is likely to occur via a tyrosine recombinase, which is conserved across phaeoviruses and contains the catalytic residues required for its function (Delaroque et al, 2001; Duchêne et al, 2025).

Xanthoviruses are a group of viruses that form a sister clade to phaeoviruses. To date, evidence for their existence comes exclusively from xanthophyte genomes carrying nucleocytoviruses hallmark viral genes, such as MCP and DNA polymerase B (Denoeud et al, 2024). Interestingly, these genomes also contain tyrosine recombinase genes that are phylogenetically sister to those of phaeoviruses, suggesting that xanthoviruses may employ the same enzymatic machinery to integrate into algal genomes and potentially share a similar latent life cycle strategy. Thus, although their sequences have diverged, the underlying integration mechanism is likely conserved across phaeoviruses and may also extend to xanthoviruses.

In contrast, punuiviruses, which may establish latent life cycles in the green algae Chlamydomonas, appear to utilize a completely different insertion mechanism (Erazo-Garcia et al, 2025). Integration of their genome results in target site duplications, a process that closely resembles transposon-mediated insertion (Fig. 2B). Their putative integrase is phylogenetically related to retroviral integrases and may be derived from virophages. Together, these comparisons highlight that while punuiviruses and phaeoviruses both belong to the Nucleocytoviricota, they may have evolved latent life cycles independently, relying on distinct enzymatic tools acquired from different evolutionary sources.

Environmental and developmental cues (e.g., temperature, onset of sexual reproduction) were suggested to break latency and activate phaeoviral EVEs. However, only recently has a clear link been established between activation of EVEs and the characteristic symptoms observed in infected Ectocarpus. Work by Duchêne et al provided this long-awaited proof, showing that reactivation of a specific EVE during reproductive development is directly associated with symptomatic phenotypes specifically in gametangia, which develop to produce viral particles instead of gametes (Duchêne et al, 2025). When transmitted to the descendance, either horizontally or vertically via the germline, these EVEs conserve their ability to reactivate. Interestingly, the phaeoviral genomes contains several sensory systems, in particular hybrid histidine kinases, that could play a role in the activation of the virus in response to developmental and environmental cues (Delaroque et al, 2001). In the Chlamydomonas-punuiviruses system, the exact trigger for viral activation remains unknown (Erazo-Garcia et al, 2025). The virus seems to activate as the culture ages, during mid-exponential to stationary phase of growth, suggesting that accumulation of metabolites or nutrient exhaustion could be triggers for viral activation. Temperature is a recurrent trigger for activation of eukaryotic viruses. For example, the mirusvirus which is found integrated in the protist Aurantiochytrium limacinum is transcribed after a cold shock experiment (Collier et al, 2023a; Chung et al, 2025). Another example is the green algae Cylindrocapsa, which produced viral particles in developing germling after a heat-shock treatment (Hoffman and Stanker, 1976). Finally, the established model for latent infection is temperate phages, which infect bacteria and remain latent in their genomes in the form of inserted prophages. Diverse environmental cues are known to trigger prophage induction, such as abiotic growth conditions (nutrients, (an)aerobic conditions, antibiotics, heavy metals, pollutants, ultraviolet radiation or temperature) but also biotic cues such as superinfection by other phages and host density (Bruce et al, 2021; Zhang et al, 2022). Whether environmental cues other than temperature are involved in phaeovirus activation remains to be investigated.

In the silent state, i.e., in the absence of viral symptoms, the inserted phaeoviruses are transcriptionally repressed, except for a handful of genes in that are constitutively expressed and could be involved in breaking the latency (Cock et al, 2010; Denoeud et al, 2024; Ban, 2025; Duchêne et al, 2025). New insights into the mechanisms involved in this repression are rising from the studies of Ectocarpus genome regulation. One method to study genome regulation is Hi-C, a genome-wide sequencing technique that captures the three-dimensional organization of chromatin by identifying physical interactions between different regions of DNA within the nucleus. Interestingly, Hi-C demonstrated that a silent EVE in Ectocarpus species 7 (strain Ec32) is sequestered within higher-order 3D chromatin domains in close proximity to telomeres, likely preventing spurious activation under normal growth conditions (Liu et al, 2024). The same EVE region is enriched in repressive histone methylation marks H3K79 and depleted of activating chromatin modifications (Bourdareau et al, 2021; Gueno et al, 2022). Chromatin remodeling is involved in repression of Nucleocytoviricota EVEs in other systems. For example, the giant EVE in the fungi Rhizophagus irregularis is in a condensed chromatin region, insulated from the rest of the chromosome (Zhao et al, 2023). Although absent in brown algae

(Vigneau et al, 2025), cytosine methylation is also recognized as a repressive mechanism to control foreign DNA. In the Discoba algae *Euglena*, cytosines within EVEs are heavily methylated, supposedly as part of the alga's antiviral defense (Jivaji et al, 2025). This pattern extends beyond: in the protist *Amoebidium appalachense*, closely related to animals, 5-methylcytosine (5mC) silences giant virus insertions (Sarre et al, 2024). These findings suggest that 5mC DNA methylation, histone modification and other chromatin-mediated repression mechanisms represent conserved eukaryotic strategies for controlling newly acquired viral DNA, potentially enabling stable coexistence between host genomes and integrated viral elements.

Herpes viruses are a well-known example of latent viruses in eukaryotes, and herpesvirus relatives (mirusviruses) are also found in the genomes of some protists (Collier et al, 2023b). Herpes simplex virus (HSV) can establish lytic infection in epithelial cells or latency in sensory neurons, with chromatin state on viral DNA regulating this switch. Herpes Human Virus 6 (HHV-6A/B) genomes integrate into telomeres of latently infected cells. Following the establishment of latent infection, viral lytic gene expression is silenced, and the lytic gene promoters are associated with repressive heterochromatin (H3K9me2/3 and H3K27me3) (Knipe and Cliffe, 2008; Cliffe et al, 2009; Kwiatkowski et al, 2009; Cliffe et al, 2013; Nicoll et al, 2016; Suzich and Cliffe, 2018). These results indicate that viral genomes reside in condensed chromatin, revealing epigenetic mechanisms underlying integrated viral genome silencing and shared principles across organisms as phylogenetically distant as brown algae and humans.

While phaeovirus activation involves changes in temperature, virtually nothing is known about the molecular mechanisms underlying this process. No connection has yet been established between chromatin-based silencing in vegetative tissues and the release of silencing in specific cell types or in response to environmental cues. The virus has been shown to exist as a circular form in symptomatic algae (Delaroque et al, 2001; Duchêne et al, 2025), but the molecular basis of provirus reactivation remains largely unexplored. The genome of EsV-1 was found to contain proteins with similarities to bacteriophage regulators of lysogeny, but this will have to be functionally validated (Delaroque et al, 2001). It also remains to be explored whether giant viruses encode proteins that counteract host antiviral inactivation mechanisms, analogous to the silencing suppressors evolved by plant DNA viruses (Burgyán and Havelda, 2011; Pumplin and Voinnet, 2013; Zhao et al, 2016; Baulcombe, 2022; Zhang et al, 2025).

The establishment of a connection between chromatin-based silencing, EVEs reactivation, and disease symptoms, three decades after the initial discovery of EsV-1, marks a major advance in understanding how EVEs shape host development.

## Ecological and evolutionary implications

The direct impact of phaeovirus on brown algal fitness is difficult to measure. Symptoms appear only in reproductive stages and no phenotypic differences could be observed between infected and healthy algae at immature stages (del Campo et al, 1997). Infected algae might produce fewer offspring, as viral replication occurs in their reproductive structures and diverts resources from gamete or spore production, though this remains to be experimentally tested. Surprisingly, several studies suggest that more than half of algal

natural populations, and in some cases up to 100% of individuals, carry viral markers in their genomes (Sengco et al, 1996; Dixon et al, 2000; Müller et al, 2000; McKeown et al, 2017; McKeown et al, 2018; Ruiz Martínez et al, 2023; Ban, 2025). This implies that the fitness cost of carrying an EVE may not be very high. However, considering that many EVEs are not active (see above, especially the examples from *Porterinema*, Mckeown et al, 2025), PCR-based detections can reflect relics of past infections rather than active ones. In fact, only ~5% of *Ectocarpus* isolates from Sengco et al (1996) and Müller et al (2000) showed symptoms in culture. Similarly, Dixon et al studied the direct occurrence of symptoms in *Ectocarpus fasciculatus* populations, with only 5% of the sampled individuals showing obvious symptoms (Dixon et al, 2000). The low prevalence of symptoms from field samples led to the idea that the virus is primarily transmitted vertically rather than by de novo infections of uninfected gametes or spores by virus particles (Sengco et al, 1996; Müller et al, 2000). The balance between vertical and horizontal transmission in field populations and whether virus may manipulate or affect host life cycle in nature remain open questions. The use of genomic data of algal populations to investigate full EVE presence coupled with demographic and epidemiological modeling of the viral life cycle could help to elucidate the preferred mode of transmission (vertical vs horizontal) of phaeoviruses in different algal species, and to predict how the virus may affect algal life cycle.

As temperature seems to be a critical parameter in the virus life cycle, this can also influence the severity of symptoms and the impact on the algal fitness. In the context of global warming, algae populations, and in particular kelps, are challenged with increasing temperature and heat waves (Ling et al, 2009; Wernberg et al, 2010; Raybaud et al, 2013; Krumhansl et al, 2016; Arafeh-Dalmau et al, 2019; Suskiewicz et al, 2024). Whether viral infections act as an additional stressor or increased temperature alleviate the stress of viral infection could influence the resilience of kelp populations, but this remains not known.

The extent to which viral integration contributes to host speciation, population structure and intraspecific variation has not been thoroughly assessed. Different *Ectocarpus* species and strains carry different phaeoviral EVEs (Denoeud et al, 2024; Duchêne et al, 2025), as do other brown algae (McKeown et al, 2017; McKeown et al, 2018; Ruiz Martínez et al, 2023; Denoeud et al, 2024; Ban, 2025), but the extent to which this shaped evolutionary dynamics at the population and species scale is unknown. Similarly, in Chlorophyta, *Nucleocytoviricota* EVEs contribute to intraspecific variability in *Chlamydomonas reinhardtii* populations, and to the evolution of green algal genomes (Moniruzzaman et al, 2020; Moniruzzaman et al, 2022), but their role in driving host evolution is yet unclear.

Viral infection has shaped the evolution of multicellular organism by providing novel genetic material. One striking example is the evolution of the placenta in placental mammals, which represents a classic case of viral gene co-option, i.e., the recruitment of viral genes by the host (Mi et al, 2000; Chuong, 2018). This leads to the question: has co-option of viral genes significantly contributed to the evolution of multicellular development in brown algae? One example of potential viral gene co-option comes from the *Ectocarpus IMMEDIATE UPRIGHT (IMM)* gene, which plays a crucial role in cell type determination during early sporophyte development in *Ectocarpus* (Macaisne et al, 2017).

The *IMM* gene encodes a protein containing repeated motifs also found in the *EsV-1-7* gene of the EsV-1. Brown algae possess large families of EsV-1-7 domain genes that are rare in other Stramenopiles, suggesting expansion of this family. Interestingly, some viral genes present in brown algal EVEs show signs of domestication such as heavy intronization or transcription without symptom production (i.e., from inactive EVEs), which highlights their potential to be co-opted by the algae (Denoeud et al, 2024; Mckeown et al, 2025). It will be necessary to examine whether these genes play a role in the development and evolution of their hosts, in order to clearly demonstrate the capacity for domesticated viral genes to transition from genomic 'passengers' to active contributors in host biology (Irwin et al, 2022).

The recent development of reverse genetic tools for brown algae (Badis et al, 2021; Martinho et al, 2025, see below) now enable to experimentally assess the roles of co-opted EVEs genes in host development and physiology. Interestingly, a substantial proportion of brown algal-specific genes are of viral origin (Barrera-Redondo et al, 2023; Denoeud et al, 2024), raising the intriguing possibility that viral integration and subsequent co-option have significantly contributed to the evolution of multicellular development in brown algae. This represents a potentially transformative mechanism whereby horizontal gene transfer from viruses has facilitated the acquisition of novel developmental capabilities, highlighting the evolutionary significance of host–virus interactions beyond traditional pathogenic contexts.

## Unlocking brown algal virology: the promise of new tools

Brown algae (Phaeophyceae), a highly diverse lineage of Stramenopiles that independently evolved complex multicellularity, exhibit a wide variety of morphologies, life cycles, and reproductive strategies, making them valuable model systems for investigating fundamental biological processes (Box 1). Over the past two decades, the brown alga *Ectocarpus* has moved from descriptive genomics into a phase where virology questions can be addressed mechanistically, as research groups worldwide have developed and adapted molecular and genetic tools (Coelho and Cock, 2020). Successive advances in genome assembly, from the first *Ectocarpus* reference genome (Cock et al, 2010) to highly contiguous chromosome-scale and near telomere-to-telomere genomes (Cormier et al, 2017; Liu et al, 2024; Barrera-Redondo et al, 2025), now allow to identify EVEs precise genomic location. These resources are complemented by transcriptomic atlases that chart the developmental program of *Ectocarpus* (Lipinska et al, 2015; Luthringer et al, 2015; Cossard et al, 2022; Lotharukpong et al, 2024; Ratchinski et al, 2025), that can be used to study expression of EVE genes across life-cycle stages. Together these tools provide a solid foundation for functional studies and a robust baseline against which viral influence can be detected and quantified.

Functional genetic tools are increasingly accessible for the brown algae. Forward genetic screens using UV or EMS mutagenesis coupled to mapping have identified regulators of development and life-cycle transitions, yielding clear genotype–phenotype links (e.g., Coelho et al, 2011; Godfroy et al, 2017; Macaisne et al, 2017; Arun et al, 2019). Reverse genetics tools have advanced rapidly. Reverse genetics initially leveraged RNA interference for targeted knockdowns, but the introduction of CRISPR-Cas9 has been a major breakthrough (Farnham et al, 2013; Badis et al, 2021;

Martinho et al, 2025). Optimized delivery by biolistic bombardment and laser-assisted microinjection introduces DNA or CRISPR-Cas9 RNPs into reproductive cells to enable precise knockouts and targeted mutagenesis. Selection using APT/2-FA exploit the haploid phase of brown algae, making recovery and phenotyping of edited lines efficient without need for backcrossing. Improved PEG-mediated RNP delivery system with CRISPR-Cas12-based genome editing method (Martinho et al, 2025) and the development of knock-in strategies should extend precise editing beyond *Ectocarpus* to additional taxa. These tools have already revealed key regulators of sex determination and sporophyte development (Luthringer et al, 2024; Martinho et al, 2025), and lay the groundwork for reporter tagging and locus-specific manipulations in *Ectocarpus* and other brown algal lineages.

Advances in transcriptomics and epigenomics have yielded temporal and regulatory insights into *Ectocarpus* development (Lipinska et al, 2015; Lotharukpong et al, 2024; Vigneau et al, 2024; Ratchinski et al, 2025). These datasets provide a quantitative framework against which viral perturbations can be detected and interpreted, supporting future efforts to identify potential viral imprints on developmental pathways and defense responses. To address the molecular mechanisms underlying the host–virus interaction, a viable possibility is low-input RNA-seq on micro-dissected, virus-affected reproductive tissues (e.g., gametangia/sporangia), enabling library construction from minute samples and tissue-level dissection of host–virus interactions in compartments where replication is concentrated. In parallel, single-cell and single-nucleus RNA-seq, already widely used in other host–virus systems, when adapted to brown algae will resolve cell-type-specific infection states and trajectories of EVE reactivation. Experience from mammalian and plant infections points to marked cell-to-cell heterogeneity, with reservoir-like cells coexisting alongside exposed yet uninfected cells, and distinct cell identities orchestrating spread or persistence (Steuerman et al, 2018; Liao et al, 2020; Tang et al, 2023; Zhu et al, 2023). In brown algae, such approaches should pinpoint when and where EVEs reactivate and which cell types permit productive replication; beyond activation states, single-cell and single-nucleus RNA-seq can delineate the transcriptional circuits, signaling modules, and cell–cell communication that mediate host–virus interplay at cellular resolution. Moreover, ChIP-seq and Hi-C offer complementary views of EVE silencing, repressive histone marks and three-dimensional chromatin landscapes (Bourdareau et al, 2021; Liu et al, 2024).

Imaging approaches are increasingly being used to localize infection in space and time (e.g., Mayer et al, 2025). Coupled with confocal microscopy, RNA fluorescence in *situ* hybridization visualizes viral or host transcripts within virus-active tissues, while immunofluorescence allows to map host pathways during antiviral responses and the subcellular distribution of viral proteins (Vincent et al, 2021; Mayer et al, 2025). Together, these approaches render the spatiotemporal dynamics of infection tractable. Electron microscopy provides ultrastructural views of virion assembly sites and infection foci (Maier et al, 2002; Aicher et al, 2021; Rigou et al, 2025), and expansion microscopy, has the potential to yield subcellular resolution in algal tissues (Gambarotto et al, 2019; Klena et al, 2023). Applied in concert, these imaging tools are expected to reveal organelle remodeling and membrane/cytoskeletal reorganization that accompany virus activation in the algal reproductive structures.

Taken together, these technical developments are expected to revolutionize brown algal virology. The final section below outlines future directions and open questions that these tools now bring within reach.

## Future directions and open questions

The expanding genetic toolkit in brown algae offers an unprecedented opportunity to dissect virus–host interactions in this lineage. Delivery methods such as biolistics, PEG-mediated transformation, and microinjection can be used to introduce viral DNA, reporter constructs, or engineered host genes directly into algal cells, enabling controlled infection assays and functional studies (Jiang et al, 2003; Zhang et al, 2008; Shen et al, 2023; Martinho et al, 2025). Forward genetic approaches, including UV or EMS mutagenesis coupled to mapping, can generate host variants with altered susceptibility or resistance to viral activation, revealing the genetic determinants of infection outcomes. The haploid life stage of *Ectocarpus* greatly facilitates such screens, although careful experimental design is required because fertility-dependent phenotypes may be delayed. Natural variation can likewise be leveraged to identify loci underlying differential susceptibility, paralleling strategies in model plants.

Reverse genetics approaches, including RNA interference and CRISPR/Cas-mediated genome editing, permit targeted knockdowns or knockouts of host genes involved in viral silencing, replication, or reactivation, and can be extended to interrogate specific viral gene functions (Farnham et al, 2013; Godfroy et al, 2017; Macaisne et al, 2017; Badis et al, 2021; Shen et al, 2023; Luthringer et al, 2024; Martinho et al, 2025). Advances toward precise knock-ins and reporter tagging on viral or host alleles, would allow direct visualization of virus–host interactions in vivo and, together with base/prime editing, enable fine-scale test of causal pathways.

Several questions now come within reach. Genomic and chromatin analyses should clarify mechanisms of integration and the nuclear positioning and chromatin states that enforce latency, and whether developmental or environmental cues remodel these states to permit activation. At the cellular level, tagged viral or host components can allow to visualize infection dynamics in real time, while natural variation and forward genetics could reveal loci controlling resistance or susceptibility. Extending these approaches beyond *Ectocarpus* to other brown algal species promises to uncover lineage-specific virus–host strategies and provide a broader understanding of viral biology in multicellular eukaryotes. Collectively, the expanding toolkit of forward and reverse genetics positions brown algae as a highly tractable system for exploring fundamental principles of virus–host interactions and the evolution of latent viral strategies. In parallel, multi-omics approaches such as hybrid genome/virome sequencing, RNA-seq at bulk and single-cell/-nucleus resolution, epigenomics, proteomics, metabolomics, and spatial imaging, provide the integrative readouts that connect these mechanisms to activation and transmission in natural populations. Together, these advances mark an exciting era for research on giant viruses and brown algae, offering unprecedented opportunities to unravel the complexities of virus–host interactions across diverse ecological and evolutionary contexts.

## Peer review information

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

   interactions using spatial and single-cell technologies. Commun Biol 6:814

## Author contributions

**Carole Duchêne**: Writing—original draft. **Liping Wang**: Writing—original draft.
**Susana M Coelho**: Writing—original draft; Writing—review and editing.

## Disclosure and competing interests statement

SC is a member of the journal's Ecology & Evolution Advisory Editorial Board.

