## [Peer Review File · The EMBO Journal]

Revisiting giant virus-host dynamics in algae: old stories and new perspectives

Carole Duchene, Liping Wang, and Susana Coelho

Corresponding author: Susana Coelho (susana.coelho@tuebingen.mpg.de)

Review Timeline:

Submission Date:	4th Oct 25
Editorial Decision:	30th Oct 25
Revision Received:	11th Nov 25
Editorial Decision:	17th Nov 25
Revision Received:	19th Nov 25
Accepted:	24th Nov 25

Editor: Yehu Moran

Transaction Report:

Dear Dr. Coelho,

Thank you for submitting your manuscript for consideration by the EMBO Journal. It has now been seen by three referees whose comments are enclosed. As you will see, all three referees express interest in your manuscript and are broadly in favour of publication, pending satisfactory minor revision.

Given the referees' positive recommendations, I would like to invite you to submit a revised version of the manuscript, addressing the comments of all three reviewers.

Please note that one of the reviewers preferred to provide their comments in a track changes format, so I attached their file to this email.

We generally allow three months as standard revision time. Yet, in light of the relatively minor nature of the comments and the time pressure due to our interest in fitting your review article into the special issue on eco-evo that is planned for early 2026, I would deeply appreciate it if you can send us your revised version within one month from today.

Please do not hesitate contacting me with any questions regarding the revision process.

Thank you for the opportunity to consider your work for publication. I look forward to your revision.

Yours sincerely,

Yehu Moran
Academic Editor
The EMBO Journal

We realize that it is difficult to revise to a specific deadline. In the interest of protecting the conceptual advance provided by the

work, we recommend a revision within 3 months (28th Jan 2026). Please discuss the revision progress ahead of this time with the editor if you require more time to complete the revisions.

Referee #1:

This review covers the history and current developments of research on host-virus interactions in the brown alga *Ectocarpus*. Even though viruses in this system were first described decades ago, little progress has been made in understanding the finer details of how virus and host co-exist and impact each other's ecology and evolution. Due to new life inhaled in this system by the reviewer's lab as well as those of others, it is certainly a good time to summarize this recent progress and put it in context with the original studies.

The review is overall well structured and written and will inform the audience on all important aspects of this virus-host system. A few comments may help to improve the manuscript even further:

1. Title: Please change "algae" to "brown algae", which reflects the content more accurately.
2. Fig. 1A: in the timeline, what do the different colors and shades mean? For the 1974 entry, change text to "first virus particles in phaeophytes observed by TEM". The next entry (1990-2001) is difficult to read due to missing punctuation or bullet points. Also, there is a typo here (paheoviral).
3. Fig. 1B needs a larger font size for the life cycle diagram. The yellow figure is better shown without the illegible figure legend.
4. Figure 1 legend contains acronyms that are not used in the figure, or again in the legend. Better introduce those at first instances in the main text. Also, there is an inconsistency: (brown arrow) vs. (blue) arrow.
5. l. 138: This "dual" lifestyle is basically a 'normal' lysogenic cycle when compared to bacteriophages, isn't it? It might be fitting to include a paragraph here on comparison with temperate phages in prokaryotes. See also the use of "distinctive" in l. 198
6. l. 170: by "structures", you mean genomes?
7. l. 174 and others: please check whether taxonomic terms need italicization (e.g. *Imitervirales*).
8. l. 212: this paragraph gets very detailed, so perhaps a short explanation of attachment sites would be good to have here.
9. l. 226: "sister" is not an adjective.
10. l. 247 and others: please define clearly what is meant by "active" and "inactive" EVEs. How can you distinguish between them?
11. l. 250: briefly explain the Hi-C method here.
12. l. 316: briefly explain co-option in this context.
13. Box 1: you may want to expand the Stramenopiles section of the tree. Even though you state in the box and legend text that brown algae belong to the Stramenopiles, the way the figure is currently laid out, it might wrongly imply to some readers that Stramenopiles are largely equivalent to brown algae. Alternatively, you could replace the tree with one that shows only the diversity of lineages within the Stramenopiles, as the other supergroups are not relevant to the topic. This could also fit with the genus names in the images below.

Referee #2:

This is a timely and interesting review of the *Ectocarpus*-virus system, which is a model for examining interactions between giant viruses and brown algae. The authors do a good job of summarizing both recent and historical findings, and putting it into context for the readers. Given the resurgence of interest in brown algae, the new genetic tools available, and the concomitant discoveries in endogenous giant viruses across eukaryotes, this work will be of great interest to a variety of readers.

I have put most comments in a text document that should be available to the authors. I made these comments together with a PhD student in my lab, who helped with the peer review. Some of these comments are more general and may not require

revision per se. Some more general comments are below.

On general comment is that I think the findings in *Ectocarpus* could be put into broader context of GEVEs in other eukaryotes a bit more. This would probably fit best in the "Genomic era" section. Most relevant citations regarding *Amoebidium*, *Euglena*, and *Rhizophagus* are already cited and discussed in a different context, but just mentioning them here to emphasize our recent and dramatic expansion in our knowledge of endogenous giant viruses across eukaryotes would be helpful. It also shows that *Ectocarpus* is a model for dynamics that are occurring across eukaryotes. An important study that focused on green algae but set the stage for much of these work is Moniruzzaman et al., *Nature*, 2020 - perhaps that work could be mentioned.

Also, the authors mention herpesviruses briefly - perhaps it would be useful to note that herpesvirus relatives (mirusviruses) are found in the genomes of some protists as well - DOI: 10.1016/j.cub.2023.10.009.

Referee #3:

The authors provide a well-balanced and comprehensive overview of brown algal virology, with appropriate emphasis on the field's historical discoveries, the subsequent slowdown in research activity, and the recent reactivation and rapid advances driven by new technologies that may further accelerate future studies. Large DNA viruses known as phaeoviruses are the main focus of this review. These viruses infect multicellular brown algae and exhibit a unique life cycle that includes a latent infection phase in the germ cell line. Phaeoviruses are notable for their ability to integrate viral genomes into host chromosomes, a feature uncommon among isolated giant viruses belonging to the phylum Nucleocytoviricota. Given the growing interest in nucleocytoviruses within the fields of virology, microbial ecology, and evolutionary biology, this review is both timely and valuable. I thoroughly enjoyed reading this well-written paper.

Major comments

I have only three major comments, which the authors may choose to consider or not.

First, the authors note that the latent phase has also been demonstrated for puiniviruses. However, in my opinion, the degree to which this has been convincingly demonstrated by the study of Maria P. Erazo-Garcia et al. (*Science*, 2025) may vary among researchers. After reading their paper, I consider it possible that their observations -- such as transcripts, viral genome sequences, and virions -- might originate from persistently infecting viruses rather than from integrated viral forms. Indeed, they identified several genotype differences between the virion-derived genome data and the integrated viral genome. They also use "cryptic" rather than "latent" in important places in their paper. If the authors of this review believe there is a distinction in the level of evidence between their own studies (e.g., Duchêne et al., 2025) and the puinivirus study, they might consider adjusting the wording accordingly. An example sentence that the authors may consider is: "The only established Nucleocytoviricota models for which a latent life cycle has been demonstrated are phaeoviruses (Duchêne et al., 2025b) and puiniviruses (Imitervirales; see Figure 2A, Erazo-Garcia et al., 2024)." I highly evaluate the work by Erazo-Garcia but the level of demonstration/proof could be important here.

Second, the authors state that "The virus has been shown to exist as a circular plasmid in symptomatic algae (Delaroque et al., 2001; Duchêne et al., 2025b)... (L.278)." I could not confirm this statement in the cited references. This sentence appears to imply the presence of an episomal form of the phaeovirus genome -- if this is indeed what the authors intend, it would be helpful to clarify it explicitly.

A glossary may also be useful if permitted by the journal. Virologists may not be familiar with the terminology describing algal life cycles (e.g., terms in Fig. 1B), while algal biologists may be less familiar with virological terms such as EVE, latent infection, and persistent (chronic) infection. It might also help to include brief explanations of experimental techniques for non-specialist readers.

Minor comments

L.20: Please provide reference(s) for the statement "Viruses are now recognized ...".

L.41: The phrase "the latent biology of Phaeoviruses remained poorly understood relative to other giant viruses" might read

more naturally without the word "latent."

L.303: "rather than" may not be necessary here.

L.323: "Host domestication" could be rephrased as "domestication by the host" or simply "domestication".

Fig. 1A (text between 1990 and 2001): "paheoviral" → "phaeoviral."

References: Duchêne et al. (2025) is listed twice; please remove the duplicate.

Because this review may serve as an entry point for many young researchers, it would be advisable to follow the ICTV rules for viral taxonomy and viral name writing conventions (<https://ictv.global/faq/names>). In short, viral taxonomic names should always be italicized, whereas virus names should be written in roman font and not capitalized (except at the beginning of a sentence). Examples of taxonomic names that should be italicized: *Phycodnaviridae*, *Pandoravirales*, *Algavirales*, *Mycodnaviridae*, *Imitervirales*, etc.

Examples of virus names that should not be italicized or capitalized: phaeoviruses, mimiviruses, chloroviruses, coccolithoviruses.

Similarly, names such as *Paramecium bursaria chlorella virus*, *Ectocarpus siliculosus virus 1*, *Prymnesium kappa virus*, and *Emiliana huxleyi virus* are virus names. These should be written in plain (non-italicized) text including the host name parts.

Referee #1:

This review cover the history and current developments of research on host-virus interactions in the brown alga *Ectocarpus*. Even though viruses in this system were first described decades ago, little progress has been made in understanding the finer details of how virus and host co-exist and impact each other's ecology and evolution. Due to new life inhaled in this system by the reviewer's lab as well as those of others, it is certainly a good time to summarize this recent progress and put it in context with the original studies.

The review is overall well structured and written and will inform the audience on all important aspects of this virus-host system. A few comments may help to improve the manuscript even further:

1. Title: Please change "algae" "brown algae", which reflects the content more accurately.

Re: this has been done

2. Fig. 1A: in the timeline, what do the different colors and shades mean? For the 1974 entry, change text to "first virus particles in phaeophytes observed by TEM". The next entry (1990-2001) is difficult to read due to missing punctuation or bullet points. Also, there is a typo here (paheoviral).

Re: Colors in the time line have been changed to match the arrows; pannel B: brown for brown algae research events, blue for Phaeoviral research events

3. Fig. 1B needs a larger font size for the life cycle diagram. The yellow figure is better shown without the illegible figure legend.

Re: This has been done

4. Figure 1 legend contains acronyms that are not used in the figure, or again in the legend. Better introduce those at first instances in the main text. Also, there is an inconsistency: (brown arrow) vs. (blue) arrow.

Re: We suppose that the reviewer refers to unilocular and plurilocular sporangia (abbreviated UL and PL). These have now been defined better in the legend, and these acronyms are used in the figure to gain space in the life cycle scheme.

Brown arrows depict the life cycle of a healthy *Ectocarpus*, while the blue arrows highlight the viral cycle.

5. l. 138: This "dual" lifestyle is basically a 'normal' lysogenic cycle when compared to bacteriophages, isn't it? It might be fitting to include a paragraph here on comparison with temperate phages in prokaryotes. See also the use of "distinctive" in l. 198

Re: Yes, the reviewer is correct: "dual" has been changed to "lysogenic". We have added a sentence on temperate phages line 502 as suggested.

A few sentences on prophage activation have also been added:

"Finally, the established model for latent infection is temperate phages, which infect bacteria and remain latent in their genomes in the form of inserted prophages. Diverse environmental cues are known to trigger prophage induction, such as abiotic growth conditions (nutrients, (an)aerobic conditions, antibiotics, heavy metals, pollutants, ultraviolet radiations or temperature) but also biotic cues such as superinfection by other phages and host density (Zhang et al, 2022; Bruce et al, 2021). Whether environmental cues other than temperature are involved in phaeoviruses activation remains to be investigated."

And we also added a sentence about induction: *“The genome of EsV-1 was found to contain proteins with similarities to bacteriophage regulators of lysogeny, but this will have to be functionally validated (Delaroque et al, 2001).”*

I.198 : the word 'distinctive' has been removed.

6. 1. 170: by "structures", you mean genomes?

Re: The reviewer is correct, “structures” has been changed to “genomes” for clarity

7. 1. 174 and others: please check whether taxonomic terms need italicization (e.g. Imitervirales).

Re: This has been corrected. Figure 2A has been modified to match the nomenclature

8. 1. 212: this paragraph gets very detailed, so perhaps a short explanation of att sites would be good to have here.

Re: This paragraph has been simplified, and for more detail the readers are directed to the Figure 2B and the reference papers.

9. 1. 226: "sister" is not an adjective.

Re: The sentence has been changed.

10. 1. 247 and others: please define clearly what is meant by "active" and "inactive" EVEs. How can you distinguish between them?

Re: thank you for the suggestion – we meant capable of activating to produce symptoms. We added more details in the corresponding sentences to increase clarity.

11. 1. 250: briefly explain the Hi-C method here.

Re: A sentence has been added to explain the HiC approach: *“genome-wide sequencing technique that captures the three-dimensional organization of chromatin by identifying physical interactions between different regions of DNA within the nucleus.”*

12. 1. 316: briefly explain co-option in this context.

Re: A small definition was added *“(…) which represents a classic case of viral gene co-option, i.e., the recruitment of viral genes by the host”*

13. Box 1: you may want to expand the Stramenopiles section of the tree. Even though you state in the box and legend text that brown algae belong to the Stramenopiles, the way the figure is currently laid out, it might wrongly imply to some readers that Stramenopiles are largely equivalent to brown algae. Alternatively, you could replace the tree with one that shows only the diversity of lineages within the Stramenopiles, as the other supergroups are not relevant to the topic. This could also fit with the genus names in the images below.

Re: We have added a schematic tree in the box depicting the major groups inside Stramenopiles and highlighting brown algae as members of Stramenopiles

Referee #2:

This is a timely and interesting review of the Ectocarpus-virus system, which is a model for examining interactions between giant viruses and brown algae. The authors do a good job of summarizing both recent and historical findings, and putting it into context for the readers. Given the resurgence of interest in brown algae, the new genetic tools available, and the concomitant discoveries in endogenous giant viruses across eukaryotes, this work will be of great interest to a variety of readers.

I have put most comments in a text document that should be available to the authors. I made these comments together with a PhD student in my lab, who helped with the peer review. Some of these comments are more general and may not require revision per se. Some more general comments are below.

One general comment is that I think the findings in Ectocarpus could be put into broader context of GEVEs in other eukaryotes a bit more. This would probably fit best in the "Genomic era" section. Most relevant citations regarding Amoebidium, Euglena, and Rhizophagus are already cited and discussed in a different context, but just mentioning them here to emphasize our recent and dramatic expansion in our knowledge of endogenous giant viruses across eukaryotes would be helpful. It also shows that Ectocarpus is a model for dynamics that are occurring across eukaryotes. An important study that focused on green algae but set the stage for much of these work is Moniruzzaman et al., Nature, 2020 - perhaps that work could be mentioned.

Re: We thank the reviewer for the positive comments and suggestions in the word document. We accepted everything, including the citations suggestions, nomenclature and language correction have also been used throughout the manuscript.

We added a new paragraph to the "genomic era" section, see also below

Also, the authors mention herpesviruses briefly - perhaps it would be useful to note that herpesvirus relatives (mirusviruses) are found in the genomes of some protists as well - DOI: 10.1016/j.cub.2023.10.009.

Re: this has been done: "*Herpes viruses are a well-known example of latent viruses in eukaryotes, and herpesvirus relatives (mirusviruses) are also found in the genomes of some protists (Collier et al, 2023)*"

L34: I would rephrase to say the latency was suggested as a plausible explanation.

Re: Thank you for the suggestion, this has been done (sentence changed to "suggested a latent life cycle with integration of the viral DNA into the host genome")

Figure 1: Perhaps some gaps could be added to make the timeline more to scale? The even spacing is a bit confusing when some events are separated by large gaps in time. Also, why are some nodes colored differently? I know the 1990-2001 is a range, so maybe coloring that green makes sense, but why is the 2017 node also green?

Re: thank you for the suggestions, we improved the figure as suggested.

Could also include GEVEs from different orders found within the same genome (eg. *Tetraselmis* GEVEs)

Re: We agree that this is indeed a valuable example. However, we prefer not to include additional references and details in the figure, as this could compromise its clarity and readability. Nevertheless, if the reviewer considers it essential, we will be happy to incorporate it.

L121-131: In this part that talks about history I would like to see a little bit more about the strains used for these studies, because the initial strain used in the beginning was not the same used when the genome was sequenced in 2010, and later on Carole added new strains. It results a little confusing connecting these results when there are differences among the host strains.

Integration of EsV-1: In just one copy and where? was this known since the beginning? I get confused because old papers indicate the virus being integrated in several fragments. I am not sure if this was just a limitation from the sequencing method and assembly

Re: We added a paragraph at the beginning of the 'genomic era' section: "*The Ectocarpus genome, and the Phaeophyceae-phaeovirus system in general, exemplifies the relationship between viral insertion and viral symptoms, as well as the dynamics of these insertions across evolutionary times. The first Ectocarpus genome to be sequenced was found to host several virus pieces, including the insertion of a putatively full length phaeovirus closely related to EsV-1 (Cock et al, 2010). However, this Ectocarpus strain (species 7, strain Ec32) is not the same strain as the one used in the historically studies on EsV-1, and never showed symptoms in culture. The Ectocarpus strain hosting EsV-1 was sequenced only later, along with other Ectocarpus strains showing symptoms, allowing the definitive demonstration that phaeoviral EVEs are reactivating to cause symptoms using classical and reverse genetics (Duchêne et al, 2025). These genomes host one or several active EVEs (i.e. able to cause symptoms), along with seemingly full-length EVEs not associated with symptoms and smaller EVE pieces, probably resulting from EVE degradation during evolution, and explaining the first reports that Ectocarpus genome contains several viral DNA pieces (Delaroque & Boland, 2008).*"

L 136: do we know more about this process?

Re: this is a very interesting topic but unfortunately the answer is no, we do not know much about the entry of phaeoviruses into their host cells. It is actually a topic we would like to explore in the future.

The genomic era section: I understand the focus on brown algae, but somewhere in this section it would be helpful to explain that a variety of other protist lineages have GEVEs as well, and how giant viruses likely shape the genome architecture across eukaryotes. This will broaden the scope of these findings and highlight how *Ectocarpus* could be useful model of more generalized processes. The *Amoebidium*, *Euglena*, and *Rhizophagus* examples cited below could be mentioned as key examples. It would also be good to mention the Moniruzzaman et al. Nature

2020 study, which was the first to describe widespread integration of giant viruses in a protist lineage (<https://doi.org/10.1038/s41586-020-2924-2>)

Re: Thank you for the suggestion, we added a paragraph introducing EVEs in diverse genomes at the beginning of the genomic era section: *“Recent advances in sequencing technologies revealed the breadth and the ubiquity of viral elements integrated in eukaryotic genomes. With more and more (good quality) eukaryotic genomes becoming available, we are only starting to realize the diversity of Nucleocytoviricota and their hosts and how these interactions shaped the evolution of eukaryotes (Moniruzzaman et al, 2022; Zhao et al, 2023; Jivaji et al, 2025; Sarre et al, 2024; Gallot-Lavallée & Blanc, 2017). However, if the detection of EVEs in a genome can inform on past encounters between a host and its viruses, the physiological and evolutionary meaning of these insertions is not always evident.”*

L181: Could it also be single endogenization events followed by duplications?

Re: Thank you for the suggestion - we added the following sentence: *“The diversity of phaeoviral EVEs and their non-congruence with brown algal phylogeny suggest repeated event of infection rather than a single event followed by duplication and diversification, but this remains to be further studied”*

L188: Would it not make sense to introduce this term earlier and use it throughout, rather than refer to EVEs above and GEVEs here?

Re: For simplicity, the reference to GEVE has been removed

L201: This generalizes too much giant virus replication strategies. Even within the lytic ones, there are marked differences, some with nuclear stages. Many NCLDV have some kind of nuclear stage. Chloroviruses are a well-studied example, but others too. Perhaps this sentence could be re-worded.

Re: The sentence has been modified to *“ phaeoviruses exhibit an integration strategy that differentiates them from many other nucleocytoviruses. Unlike typical giant viruses such as mimiviruses, which exhibit primarily lytic replication strategies (Scola et al, 2003; Machado et al, 2022), phaeoviruses establish long-term latent infections..”*

L208-209: This idea has been brought up a couple times. Perhaps to avoid redundancy I would add some examples to give it more support. For example: Moss viral genes transcription (<https://onlinelibrary.wiley.com/doi/full/10.1111/tpj.13801>), Chlamy intraspecies genomic variability (Endogenous giant viruses contribute to intraspecies genomic variability in the model green alga *Chlamydomonas reinhardtii*), amoebidium etc. I think this material could be discussed in more depth in the previous section – that way the “impacts on genome host evolution” is there, and this section could be more focused on mechanisms.

Re: Thank you. This idea has been moved to the evolutionary implication sections, with a sentence describing the situation in green algae, discussed in the light of population genetics and speciation, before going to evolutive innovation in the following paragraph.

*“In the case of brown algae, the viral integration not only facilitates viral persistence but also contributes to host evolution, potentially influencing genome architecture in ways uncommon among giant viruses. The extent to which viral integration contributes to host speciation, population structure and intraspecific variation has not been thoroughly assessed. Different *Ectocarpus* species and strains carry different phaeoviral EVEs (Denoeud et al, 2024; Duchêne et al, 2025), as do other brown algae (Denoeud et al, 2024; McKeown et al, 2017, 2018; Ruiz Martínez et al, 2023; Ban, 2025), but the extent to which this*

shaped evolutionary dynamics at the population and species scale is unknown. Similarly, in Chlorophyta, Nucleocytoviricota EVEs contribute to intraspecific variability in Chlamydomonas reinhardtii populations, and to the evolution of green algal genomes (Moniruzzaman et al, 2022, 2020), but their role in driving host evolution is yet unclear. However, examples emerge as to how viruses contributed to key transitions in the evolution of brown algae.”

L249: Would be nice to have a small section on other plausible mechanisms for induction. Cold stress in mirusviruses, or other cues suggested by old VLP papers.

Re: This is a good idea, we have added a few sentences: “*Temperature is a recurrent trigger for activation of eukaryotic viruses. For example, the mirusvirus which is found integrated in the protist Aurantiochytrium limacinum transcribed after a cold shock experiment (Chung et al, 2025; Collier et al, 2023b). Another example is the green algae Cyllindrocapsa, which produced viral particles in developing germling after a heat-shock treatment (Hoffman & Stanker, 1976).*”

L292: would avoid strong terms like this. Perhaps re-word to “have been difficult to measure” or something similar?

Re: The sentence has been re-worded

L304 This raises an interesting circumstance, in which perhaps we don't know yet which phenotypes could be viral-related. These could be hard to identify. Old VLP papers also report a very low frequency of giant VLPs in field samples of many types of algae, and many of those were associated with specific cell stages (frequently those related to a sexual stage) within the species.

Re: It is indeed very interesting, and indeed we do not have yet a clear idea of which phenotypes are a direct consequence of viral infection. Note that the first observations of virus-like particles in brown algae are from field samples (Pylaiella, Markey 1973 and Ectocarpus, Clitheroe and Evans 1974). However, the frequency of individuals with viral symptoms in field populations is not yet fully described.

L317: Is known if viral infection is a problem for kelp populations in nature?

Re: This is not known to date, because fitness measurements are not easy to address in brown algae, and we do not have yet a good description of frequency of ‘active’ virus in field populations.

Referee #3:

The authors provide a well-balanced and comprehensive overview of brown algal virology, with appropriate emphasis on the field's historical discoveries, the subsequent slowdown in research activity, and the recent reactivation and rapid advances driven by new technologies that may further accelerate future studies. Large DNA viruses known as phaeoviruses are the main focus of this review. These viruses infect multicellular brown algae and exhibit a unique life cycle that includes a latent infection phase in the germ cell line. Phaeoviruses are notable for their ability to integrate viral genomes into host chromosomes, a feature uncommon among isolated giant viruses belonging to the phylum Nucleocytoviricota. Given the growing interest in

nucleocytoviruses within the fields of virology, microbial ecology, and evolutionary biology, this review is both timely and valuable. I thoroughly enjoyed reading this well-written paper.

Major comments

I have only three major comments, which the authors may choose to consider or not.

First, the authors note that the latent phase has also been demonstrated for punuiviruses. However, in my opinion, the degree to which this has been convincingly demonstrated by the study of Maria P. Erazo-Garcia et al. (Science, 2025) may vary among researchers. After reading their paper, I consider it possible that their observations -- such as transcripts, viral genome sequences, and virions -- might originate from persistently infecting viruses rather than from integrated viral forms. Indeed, they identified several genotype differences between the virion-derived genome data and the integrated viral genome. They also use "cryptic" rather than "latent" in important places in their paper. If the authors of this review believe there is a distinction in the level of evidence between their own studies (e.g., Duchêne et al., 2025) and the punuivirus study, they might consider adjusting the wording accordingly. An example sentence that the authors may consider is: "The only established Nucleocytoviricota models for which a latent life cycle has been demonstrated are phaeoviruses (Duchêne et al., 2025b) and punuiviruses (Imitervirales; see Figure 2A, Erazo-Garcia et al., 2024)." I highly evaluate the work by Erazo-Garcia but the level of demonstration/proof could be important here.

Re: Thank you for your positive comments on our manuscript. We slightly changed the wording when referring to the study of Erazo-Garcia et al, 2024.

Fig2 legend: "viral lineages for which a latent life cycle is demonstrated (phaeoviruses) or strongly suspected (punuiviruses)."

L 329: in punuiviruses (*Imitervirales*, see **Figure 2A**, Erazo-Garcia et al., 2024) a cryptic infection appears to correspond to *bone fide* latency

L351: In contrast, punuiviruses, which putatively establish latent life cycles

Second, the authors state that "The virus has been shown to exist as a circular plasmid in symptomatic algae (Delaroque et al., 2001; Duchêne et al., 2025b)... (L.278)." I could not confirm this statement in the cited references. This sentence appears to imply the presence of an episomal form of the phaeovirus genome -- if this is indeed what the authors intend, it would be helpful to clarify it explicitly.

Re: We apologize for the wrong wording, we meant a circular form that can be detected when algae are symptomatic, probably from replicating viruses or mature virions DNA. The word "plasmid" has been changed to "circular form"

A glossary may also be useful if permitted by the journal. Virologists may not be familiar with

the terminology describing algal life cycles (e.g., terms in Fig. 1B), while algal biologists may be less familiar with virological terms such as EVE, latent infection, and persistent (chronic) infection. It might also help to include brief explanations of experimental techniques for non-specialist readers.

Re: a glossary has been added

Minor comments

L.20: Please provide reference(s) for the statement "Viruses are now recognized ...".

Re: this has been done (Suttle, 2005, *Viruses in the sea*)

L.41: The phrase "the latent biology of Phaeoviruses remained poorly understood relative to other giant viruses" might read more naturally without the word "latent."

Re: done

L.303: "rather than" may not be necessary here.

Re: Perhaps we misunderstood the reviewer's point, but we would prefer to keep the phrasing because we are opposing two modes of transmission, therefore we believe "rather than" is appropriate: "*the idea that the virus is primarily transmitted vertically rather than by de novo infections of uninfected gametes or spores*"

L.323: "Host domestication" could be rephrased as "domestication by the host" or simply "domestication".

Re: done

Fig. 1A (text between 1990 and 2001): "paheoviral" → "phaeoviral."

Re: done

References: Duchêne et al. (2025) is listed twice; please remove the duplicate.

Re: done

Because this review may serve as an entry point for many young researchers, it would be advisable to follow the ICTV rules for viral taxonomy and viral name writing conventions (<https://ictv.global/faq/names>). In short, viral taxonomic names should always be italicized, whereas virus names should be written in roman font and not capitalized (except at the beginning of a sentence).

Examples of taxonomic names that should be italicized: Phycodnaviridae, Pandoravirales, Algavirales, Mycodnaviridae, Imitervirales, etc.

Examples of virus names that should not be italicized or capitalized: phaeoviruses, mimiviruses,

chloroviruses, coccolithoviruses.

Similarly, names such as *Paramecium bursaria chlorella virus*, *Ectocarpus siliculosus virus 1*, *Prymnesium kappa virus*, and *Emiliana huxleyi virus* are virus names. These should be written in plain (non-italicized) text including the host name parts.

Re: This has also been noted by the other reviewers and has been corrected throughout the manuscript. Similarly, viral names in Figure 2 have been corrected.

Dear Dr. Coelho,

Thank you for submitting your manuscript for consideration by the EMBO Journal. Based on your revision and response letter I believe that the paper has considerably improved and is close to acceptance. Yet, there are several edits which are required from technical and styling point of view:

Technical issues (raised by editorial assistance team)

While technical in nature, these must be corrected before we can proceed with official acceptance to The EMBO Journal.

- MANUSCRIPT FORMAT: please remove the figures from the manuscript text; they should only be uploaded as separate high res figure files
- Keywords: missing, please add up to five.
- ACKNOWLEDGEMENTS/FUNDING: nothing was included in the manuscript text & our system. Please check.
- DISCLOSURE AND COMPETING INTERESTS STATEMENT: missing, please add. Please include a sentence about Dr. Coelho being on the editorial board of the journal.
- FIGURE LEGENDS: Please move them to the end of the manuscript text.

styling issues/suggestions

These were made by me (Dr. Yehu Moran) and should be treated as mere suggestions (can be ignored if I got something wrong). Yet, please note such edits are barely made (if at all) at the proof stage by Springer-Nature, and we do not tend to count on them doing it properly for Review articles, so now is the right time to make such styling edits.

1. Small typos:

- "viruses ... displays remarkable diversity". Should be "display".
- "virus-like particles was first visualized". Should be "were first visualized".
- "double stranded DNA". Should be "double-stranded DNA".
- "mid- exponential". Should be "mid-exponential".
- "cell-wall free". Should be "cell-wall-free".
- "meispore". Maybe should be "meiospore"?
- "carying" should be "carrying".
- "transformation" should be "transformation".
- "ChipSeq" should be "ChIP-seq".

2. Flow issues:

- Repetition of the Ectocarpus life cycle in multiple sections. Please make sure it is necessary in each place it appears.
- Repetition of 'active vs inactive EVEs' concept.
- Evolutionary section jumps between unrelated topics and some restructuring might improve clarity.
- The integration mechanism section is a bit hard to follow. It might be worth streamlining a bit if possible.
- Some sentences are very long and could be split for clarity.

Thank you for the opportunity to consider your work for publication. I look forward to your revision so we can proceed with official acceptance.

Yours sincerely,

Yehu Moran
Academic Editor
The EMBO Journal

We realize that it is difficult to revise to a specific deadline. In the interest of protecting the conceptual advance provided by the work, we recommend a revision within 3 months (15th Feb 2026). Please discuss the revision progress ahead of this time with the editor if you require more time to complete the revisions.

The authors addressed the remaining editorial issues.

Dear Dr. Coelho,

I am very glad to inform you that your manuscript has been accepted for publication in the EMBO Journal.

Yours sincerely,

Yehu Moran
Academic Editor
The EMBO Journal

Please note that it is The EMBO Journal policy for the transcript of the editorial process (containing referee reports and your response letters) to be published as an online supplement to each paper. If you should prefer removal of any referee-only figures included in the point-by-point response(s), e.g. because they may still be used for future publication or because they have been reproduced from published work by others, please do let us know immediately via response email.

More information is available here: https://www.embopress.org/transparent-process#Review_Process
